# Metallothionein 3 Promotes Osteoblast Differentiation in C2C12 Cells via Reduction of Oxidative Stress

**DOI:** 10.3390/ijms22094312

**Published:** 2021-04-21

**Authors:** Santie Li, Myeong-Ji Kim, Sung-Ho Lee, Litai Jin, Weitao Cong, Hye-Gwang Jeong, Kwang-Youl Lee

**Affiliations:** 1College of Pharmacy and Research Institute of Pharmaceutical Sciences, Chonnam National University, Gwangju 61186, Korea; lisantie333@163.com (S.L.); audwlekt@naver.com (M.-J.K.); puzim23@gmail.com (S.-H.L.); 2School of Pharmaceutical Science, Wenzhou Medical University, Wenzhou 325000, China; jin_litai@126.com (L.J.); cwt97126@126.com (W.C.); 3College of Pharmacy, Chungnam National University, Daejeon 34134, Korea

**Keywords:** metallothinnein 3, osteoblast differentiation, oxidative stress, runt-related gene 2, osterix, distal-less homeobox 5

## Abstract

Metallothioneins (MTs) are intracellular cysteine-rich proteins, and their expressions are enhanced under stress conditions. MTs are recognized as having the ability to regulate redox balance in living organisms; however, their role in regulating osteoblast differentiation is still unclear. In this research, we found that the expression of MT3, one member of the MT protein family, was specifically upregulated in the differentiation process of C2C12 myoblasts treated with bone morphogenetic protein 4 (BMP4). Transfection with MT3-overexpressing plasmids in C2C12 cells enhanced their differentiation to osteoblasts, together with upregulating the protein expression of bone specific transcription factors runt-related gene 2 (Runx2), Osterix, and distal-less homeobox 5 (Dlx5). Additionally, MT3 knockdown performed the opposite. Further studies revealed that overexpression of MT3 decreased reactive oxygen species (ROS) production in C2C12 cells treated with BMP4, and MT3 silencing enhanced ROS production. Treating C2C12 cells with antioxidant N-acetylcysteine also promoted osteoblast differentiation, and upregulated Runx2/Osterix/Dlx5, while ROS generator antimycin A treatment performed the opposite. Finally, antimycin A treatment inhibited osteoblast differentiation and Runx2/Osterix/Dlx5 expression in MT3-overexpressing C2C12 cells. These findings identify the role of MT3 in osteoblast differentiation and indicate that MT3 may have interesting potential in the field of osteogenesis research.

## 1. Introduction

Osteoblast differentiation is a process involved in bone tissue self-renewal throughout the period of a whole lifetime, and its dysfunction leads to skeletal disorders such as osteoporosis [1], a disease caused by systemic bone mass reduction [2]. Mesenchymal stem cells and skeletal muscle cells play important roles in bone homeostasis maintenance, as they can differentiate into osteoblastic-like cells and thus affect bone remodeling [3,4]. During this process, several signaling pathways such as bone morphogenetic protein (BMP), Wnt, Notch, and Hedgehog all contribute greatly to the differentiation of osteoblasts [5]. In the current treatment of bone mass loss related diseases, bone forming drugs and anti-bone resorptive drug are widely used; however, these drugs show serious side effects after long term clinical use [6]. Therefore, effective therapeutic strategies targeting osteoblast differentiation with fewer side effects will contribute greatly to the control of osteogenesis-related disorders.

The metallothionein (MT) protein family consists of four main isoforms (MT1–MT4), all of which are low molecular weight, cysteine-rich, metal-binding proteins that perform a variety of functions in the control of body homeostasis [7]. MTs are traditionally considered to be intracellular proteins which are expressed in various kinds of cells and tissues. They play essential roles in heavy metal detoxification, cellular redox balance regulation, immunomodulation, and cell proliferation [7,8]. In the field of osteogenesis, MT has been shown to have the ability to protect against hydrogen peroxide-induced inhibition of osteoblast differentiation in mouse bone marrow stromal cells, and a zinc diet (which will lead to endogenous MT induction) or supplementation can also influence osteoblast differentiation, as well as postnatal bone growth [9,10]. These studies indicate the potential of the MT protein family for osteoblast differentiation; however, the role and detailed mechanisms of the specific isoforms of MT in osteoblast differentiation are still unclear. Interestingly, in some clinical-related studies, researchers have defined that osteoblast gene expression is crucial to areas reconstructed or treated with biomaterials [11,12,13,14], or in areas treated with sinus lifting [15,16,17]. As intracellular proteins, MTs can also be considered as potential targets in these fields, which makes it meaningful to identify the role of specific isoforms of MT in regulating osteoblast differentiation.

In the regulation of osteoblast differentiation, several transcription factors such as runt-related gene 2 (Runx2), Osterix, and distal-less homeobox 5 (Dlx5) are essential to mediate the expression of osteoblast related genes such as type I collagen, bone sialoprotein, and Osteopontin [18]. Runx2 is the first described transcription factor to regulate osteoblast differentiation, and the knockout of Runx2 in mice has led to osteoblast loss and bone dysplasia [19]. Osterix is considered to be the downstream target of Runx2, and it also acts as a zinc finger-containing transcription factor, which plays important roles in osteoblast differentiation [20]. For Dlx5, a transcription factor which is expressed in the early stages of bone synthesis [18], its expression supposedly has a close relationship with osteoblast differentiation and bone cell phenotype maturation [21]. All of these three factors work together to influence the process of osteogenesis, and their activities are also intricately connected [22].

Oxidative stress is defined as an imbalance between reactive oxygen species (ROS) production and antioxidant defenses in living cells and organs, which is closely related with different kinds of tissue injury [23]. The overproduction of ROS may also influence cellular functions in different pathological conditions, including osteoporosis [24]. In vascular smooth muscle cells, oxidative stress enhances osteoblastic differentiation, while in bone marrow stromal cells, hydrogen peroxide treatment decreases the expression of differentiation markers of bone osteoblastic cells [25]. Several studies have also shown that elevated levels of ROS could inhibit osteoblast differentiation in both MC3T3-E1 cells (an osteogenic cell line derived from newborn mouse calvaria) and primary osteoblasts [26,27]. Interestingly, MTs have also been recognized to be powerful antioxidative factors [28], which suggests that MTs are closely related to osteogenesis. However, the relationship between intracellular MT and osteoblast differentiation has never been realized in previous studies.

In this study, we first analyzed the expression of different isoforms of MT in mouse myoblast C2C12 cells treated with or without BMP4, and found MT3 was specifically upregulated during the differentiation process. Further experiments using MT3-overexpressing plasmids and MT3-interfering RNAs showed that MT3 could upregulate Runx2/Osterix/Dlx5 expression and promote osteoblast differentiation in C2C12 cells, and this capacity of MT3 was largely dependent on its ability to scavenge ROS. Our study indicates the potential of MT3 for osteoblast differentiation, and suggests that MT3 can be used as an interesting target in the research field of osteogenesis.

## 2. Results

### 2.1. MT3 Is Significantly Upregulated in BMP4 Treated C2C12 Cells

The MT protein family consists of four main isoforms; among them, MT4 is considered to be specifically expressed in the epithelial tissue [29]. Therefore, in order to test the correlation between MTs and osteoblast differentiation, we first used quantitative real-time PCR to analyze the expression levels of MT1, MT2, and MT3 in C2C12 cells treated with or without BMP4. As shown in Figure 1A, we found that only MT3 was significantly upregulated after BMP4 treatment for 72 h in C2C12 cells. To further confirm this result, we used western blotting to analyze the protein expression levels of MT1, MT2 and MT3 in our model. The results showed that only MT3 had increased, along with the upregulation of Alkaline phosphatase (ALP), a specific marker of osteoblast differentiation, while MT1 and MT2 did not show any significant change during this process (Figure 1B). Together, these results confirmed that intracellular MT3 was specifically upregulated during osteoblast differentiation.

### 2.2. MT3 Overexpression Promotes BMP4-Induced Osteoblast Differentiation in C2C12 Cells

To identify the role of MT3 in osteoblast differentiation, we first used plasmid transfection to achieve MT3 overexpression in C2C12 cells. After successful transfection (Appendix A), ALP staining showed that MT3 overexpression promoted osteoblast differentiation induced by BMP4 (Figure 2A), which means that MT3 may play a positive role in this process. As Runx2, Osterix, and Dlx5 are important markers and regulators of osteoblast differentiation, we also used western blotting to analyze their protein expression levels in our model, and the results showed that MT3 overexpression enhanced the expression of Runx2/Osterix/Dlx5 during osteoblast differentiation (Figure 2B). Interestingly, we found BMP4-induced upregulation of phosphorylated Smad1/5 was not changed upon MT3 overexpression (Figure 2B), indicating that MT3 did not directly affect BMP signaling in our model.

### 2.3. MT3 Knockdown Inhibited BMP4-Induced Osteoblast Differentiation in C2C12 Cells

To further confirm that intracellular MT3 can regulate the process of osteoblast differentiation, we used small interfering RNAs to achieve MT3 knockdown in C2C12 cells (Appendix A), and ALP staining showed that BMP4-induced osteoblast differentiation was inhibited in those cells (Figure 3A). Furthermore, compared to control cells treated with BMP4, the protein expression levels of Runx2, Osterix, and Dlx5 were also downregulated in MT3 knockdown C2C12 cells (Figure 3B), which means that intracellular MT3 is an important regulator of osteoblast differentiation. Similarly, BMP4-induced upregulation of p-Smad1/5 was not changed upon MT3 knockdown (Figure 3B). Together, these results concluded that MT3 promoted osteoblast differentiation and upregulated Runx2/Osterix/Dlx5 expression during this process.

### 2.4. MT3 Indirectly Regulates Runx2/Osterix/Dlx5 Activation during Osteoblast Differentiation

During the process of osteoblast differentiation, Runx2, Osterix, and Dlx5 are important transcription factors which regulate the expression of osteoblastic-related genes [11], and multiple upstream factors can also influence the expression and activation of Runx2/Osterix/Dlx5 [30,31]. To further analyze the relationship between MT3 and Runx2/Osterix/Dlx5, we first used a luciferase reporter assay to test the transcriptional activities of Runx2, Osterix, and Dlx5 in 293T cells transfected with or without MT3-overexpressing plasmids. We found that MT3 overexpression had no effects on the indicated reporter activities (Figure 4A). However, in C2C12 cells treated with BMP4, MT3 overexpression significantly upregulated the luciferase reporter activities of Runx2, Osterix, and Dlx5 (Figure 4B). Moreover, MT3 knockdown in 293T cells also had no effects on the transcriptional activities of Runx2/Osterix/Dlx5 as indicated by the luciferase reporter assays (Figure 4C), but in C2C12 cells treated with BMP4, MT3 knockdown significantly inhibited the upregulation of Runx2, Osterix, and Dlx5 reporter activities (Figure 4D). Together, these results suggested that MT3 indirectly upregulated the transcriptional activities of Runx2, Osterix, and Dlx5 during osteoblast differentiation.

### 2.5. MT3 Reduces Oxidative Stress in BMP4 Treated C2C12 Cells

Given that MT3 could upregulate the protein expression of Runx2, Osterix, and Dlx5 during osteoblast differentiation, but indirectly regulated their transcriptional activities, we sought to determine how MT3 directly influences osteoblast differentiation in C2C12 cells. As the MT protein family is commonly considered to have powerful antioxidant capacities [28], and oxidative stress also highly influences osteoblast differentiation [24], we considered whether MT3 could regulate ROS generation in our model. To test this hypothesis, we used 2′,7′-dichlorofluorescin diacetate (DCFH-DA) staining and Dihydroethidium (DHE) staining to measure the ROS levels in MT3-overexpressing and MT3-silencing C2C12 cells. The results showed that BMP4 treatment highly induced ROS production, while MT3 overexpression inhibited the generation of ROS, and MT3 knockdown accelerated ROS production (Figure 5A–D), which indicated that MT3 played a positive role in reducing oxidative stress during BMP4-induced osteoblast differentiation.

### 2.6. Oxidative Stress Regulates BMP4-Induced Osteoblast Differentiation and Affects the Reporter Activities of Runx2/Osterix/Dlx5 in MT3-Overexpressing/Silencing C2C12 Cells

To fully understand the role and mechanisms of MT3 in osteoblast differentiation, and to determine that ROS overproduction inhibits osteoblast differentiation in our model, we utilized the following experiments. Firstly, we added N-acetylcysteine (NAC), a well-known antioxidant, to the culture medium of C2C12 cells, and found that BMP4-induced ROS generation was significantly inhibited (Appendix A). In the meantime, BMP4-induced osteoblast differentiation was also enhanced upon NAC treatment, as indicated by ALP staining (Appendix A), as well as the protein expression of Runx2, Osterix, and Dlx5 (Appendix A). We then treated C2C12 cells with antimycin A, an ROS generator, and found that the overproduction of ROS (Appendix A) inhibited osteoblast differentiation in our model, as indicated by ALP staining and measurement of Runx2/Osterix/Dlx5 expression (Appendix A). These results showed that oxidative stress played a negative role in the process of osteoblast differentiation in our model.

We then used luciferase reporter assays to test the effect of NAC on the transcriptional activities of Runx2/Osterix/Dlx5 in MT3-silencing C2C12 cells, and the results indicated that NAC treatment restored the inhibitory effects of the knockdown of MT3 on the transcriptional activities of Runx2, Osterix, and Dlx5 (Figure 6A). Similarly, antimycin A treatment suppressed the effects of the overexpression of MT3 on the transcriptional activity of Runx2, Osterix, and Dlx5 (Figure 6B). These results indicated that the ability of MT3 to regulate the reporter activities of Runx2/Osterix/Dlx5 was dependent on its function to reduce oxidative stress.

### 2.7. ROS Production Impedes BMP4-Induced Osteoblast Differentiation in MT3-Overexpressing C2C12 Cells

Considering the above results, we hypothesized that MT3 regulated ROS production and thus affected osteoblast differentiation in our model. To verify this hypothesis, we treated MT3-overexpressing C2C12 cells with or without antimycin A. The results indicated that antimycin A treatment induced overproduction of ROS in the presence of MT3 overexpression (Figure 7A,B), which also significantly inhibited the enhanced osteoblast differentiation mediated by MT3 overexpression (Figure 7C), as well as decreased the protein expression of Runx2, Osterix, and Dlx5 (Figure 7D). These results showed that the generation of ROS could impede the effects of MT3 to promote osteoblast differentiation, which suggested that the function of MT3 to regulate osteoblast differentiation in our model was largely dependent on its ability to reduce oxidative stress.

## 3. Discussion

In this present study, we show that intracellular MT3 is significantly upregulated in the process of osteoblast differentiation. As a member of the MT protein family, MT3 is commonly considered to be expressed in the nervous system [32,33]. It has been shown to have growth inhibitory effect in the brain [34], with powerful antioxidant capacities [35]. Our study suggests that MT3 is a novel factor to enhance osteoblast differentiation; MT3 regulates ROS production in this process, and promotes the protein expression of Runx2, Osterix, and Dlx5 in C2C12 cells treated with BMP4. Moreover, we also showed that the function of MT3 to regulate osteoblast differentiation is largely dependent on its capacity to reduce oxidative stress. In agreement with our study, zinc supplementation, which will induce intracellular MT expression, has also been proven to have the ability to promote osteoblast differentiation [10,36].

Many possible interactions between oxidative stress and osteoblast differentiation have been identified [37,38]. Our data identifies that the process of BMP4-induced osteoblast differentiation is associated with massive ROS production, which clearly indicates a state of oxidative stress. By using the antioxidant NAC and the ROS generator antimycin A, we show that oxidative stress plays a negative role in the regulation of osteoblast differentiation in our model, which means that antioxidant therapy may act as a useful strategy in future clinical studies. Interestingly, MT3 has been shown to have powerful ROS eliminating abilities in our study, which indicates its potential for future translational medicine studies in the field of osteogenesis research. More importantly, by using MT3 knockdown experiments, we also showed that intracellular MT3 expression is important for maintaining the ability of osteoblast differentiation in C2C12 cells, which means MT3 may serve as an internal regulator in bone homeostasis maintenance. On the other hand, a previous report has also indicated that the generation of ROS could promote BMP2-induced osteoblast differentiation in 2T3 pre-osteoblasts [39], which exerts an opposite result to our model. Although this report studies osteoblast differentiation in different models using different methods, it still reminds us that the effect of MT3 may be complicated in diverse tissues or cells.

Runx2, Osterix, and Dlx5 are essential transcription factors to mediate the process of osteoblast differentiation [18,40], and they are also key factors induced by BMP signaling [41]. The transcriptional targets of the three factors are all crucial proteins for osteoblast differentiation and associated bone development [42]. In our study, BMP4 induced osteoblast differentiation is associated with the upregulation of Runx2, Osterix, and Dlx5 in C2C12 cell, and MT3 overexpression or knockdown also significantly influences the expression of these factors during the differentiation process. It should be noted that our data indicate MT3 cannot directly regulate the transcriptional activity of Runx2/Osterix/Dlx5, which indicates that MT3 controls osteoblast differentiation (largely by regulating ROS production) and thus influences the expression of Runx2, Osterix, and Dlx5.

Osteoblast differentiation is also associated with the development of oral diseases; for example, bone regeneration highly influences the progression of periodontal disease [43], and the ability to promote osteoblast differentiation is necessary for the design of dental implants [44]. In infection conditions, some studies have suggested that diminished rates of bacterial colonization will significantly improve the success and survival of implant-prosthetic rehabilitations in immunocompromised patients [45,46,47] and also in endodontics [48]. It may also avoid facial peri mandibular abscesses [49]. As an intracellular factor which regulates the differentiation of osteoblastic cells, it will be interesting for future studies to analyze the role of MT3 on osteoblasts in infection conditions, which could further identify the potential clinical advantages of MT3.

Another interesting result is that our study showed BMP4 treatment induced the upregulation of MT3 in C2C12 cells, but not MT1 or MT2. Considering that previous studies have suggested that zinc finger transcription factor Sp1 increases the promoter activity of MT3 [28,50,51], and Osterix (Sp7) can act at the canonical Sp1 DNA-binding sites to regulate gene transcription [52], together with the result that BMP4 treatment can upregulate Osterix expression, it can be supposed that the increased expression of MT3 may act as a direct effect of BMP signaling in our model.

It should be noted that beyond the capacity of reducing oxidative stress, MT protein family members have long been considered to be transition metal-binding proteins that can significantly affect metal homeostasis in living organisms [53]. Although the ability of MT3 to regulate ROS production is closely associated with its metal-binding affinity [35], indicating experiments about the relationship between MT3 and metal homeostasis in the differentiation process of osteoblasts should be performed in future studies. In addition, as MT1 and MT2 can also affect ROS levels in different organs and tissues [54], it will be interesting to analyze whether exogenous MT1 or MT2 has the ability to promote osteoblast differentiation in skeletal muscle cells or mesenchymal stromal cells. Furthermore, in vivo studies of MT3 in the regulation of osteoblast differentiation will be needed to further validate the potential of MT3 to protect against bone-related disorders such as osteoporosis. It is also worth noting that in this study we did not use a large number of samples in each experiment; this clearly does not blunt the statistical significance of differences and reaching conclusions, but shall be considered as a limitation of our research.

In summary, our data demonstrate the promotive effects of intracellular MT3 in osteoblast differentiation. MT3 acts as a powerful antioxidant and reduces oxidative stress in the process of osteoblast differentiation. MT3 regulates the expression of Runx2/Osterix/Dlx5, and its ability to promote osteoblast differentiation was achieved largely by reducing oxidative stress. Our findings demonstrate the positive role of MT3 in the field of osteogenesis, and show some of the potential MT3 may have in future studies toward clinical evaluation.

## 4. Materials and Methods

### 4.1. Cell Culture and In Vitro Induction of Osteoblast Differentiation

Mouse C2C12 myoblast cell lines and human embryonic kidney 293T cell lines obtained from the American Type Culture Collection were both maintained at 37 °C in a humidified incubator with 5% CO_2_ and 95% air. Cells were cultured in Dulbecco’s modified Eagle’s medium (Gibco, Carlsbad, CA, USA) supplemented with 10% fetal bovine serum (Gibco) and 1% penicillin-streptomycin (Gibco). C2C12 cells were treated with 50 ng/mL BMP4 in the culture medium for indicated time points to induce osteoblast differentiation.

### 4.2. Western Blotting

C2C12 cells were lysed with an ice-cold radio immunoprecipitation assay lysis buffer containing protease and phosphatase inhibitor cocktails (Abcam, Cambridge, UK). Protein lysates were cleared by centrifugation at 4 °C for 15 min, and the supernatants were collected for further experiments. 30 μg total protein samples (each lane) mixed with loading buffer (Beyotime) were subjected to sodium dodecyl sulphate-polyacrylamide gel electrophoresis and then transferred to polyvinylidene fluoride membrane (Millipore, Burlington, MA, USA). Primary antibodies used to detect specific proteins were: MT1/2 (Abcam, ab95042; 1:1000 dilution), MT3 (Santa Cruz Biotechnology, Santa Cruz, CA, USA, sc-164990; 1:500 dilution), ALP (Abcam, ab229126; 1:1000 dilution), Runx2 (Abcam, ab236639; 1:1000 dilution), Osterix (Abcam, ab229258; 1:2000 dilution), Dlx5 (Abcam, ab109737; 1:3000 dilution), p-Smad1/5 (Cell Signaling Technology, Danvers, MA, USA, 9516; 1:1000 dilution), and GAPDH (Abcam, ab8245, 1:5000 dilution). After incubation with appropriate secondary antibodies, protein bands were visualized by using electrochemiluminescence reagent (Millipore), and images were captured by Amersham Image 600 system (GE Healthcare Life Sciences, Marlborough, MA, USA).

### 4.3. RNA Isolation and Quantitative Real-Time PCR

TRIzol reagent (Invitrogen, Carlsbad, CA, USA) was used to extract total RNA from C2C12 cells. Reverse transcript cDNA was established by using the GoScript Reverse Transcription System (Promega, Madison, WI, USA), and an SYBR Green PCR Assay Kit (Thermo Fisher Scientific, Waltham, MA, USA) was used to perform quantitative real-time PCR following the manufacturers’ protocols. All of the mRNA levels were normalized to GAPDH expression. Primers used in this experiment were: MT1 (Forward: AAGAGTGAGTTGGGACACCTT; Reverse: CGAGACAATACAATGGCCTCC); MT2 (Forward: ATGCAAATGTACTTCCTGCAAGA; Reverse: CTGGGAGCACTTCGCACAG); MT3 (Forward: TGCACCTGCTCGGACAAAT; Reverse: CCTTGGCACACTTCTCACATC); GAPDH (Forward: AATGGATTTGGACGCATTGGT; Reverse: TTTGCACTGGTACGTGTTGAT).

### 4.4. MT3 Overexpression and Knockdown in C2C12 Cells

The flag-tagged MT3 expression plasmids (constructed in a CMV promoter-derived mammalian expression vector) were used to mediate MT3 overexpression, and small interfering RNAs containing mouse MT3 silencing sequences (Forward: CCAAGGACUGUGUGUGCAATT; Reverse: UUGCACACACAGUCCUUGGTT) were used to mediate MT3 knockdown in C2C12 cells. Both overexpression and knockdown experiments were achieved by using the Lipofectamine 2000 transfection reagent (Thermo Fisher Scientific, Waltham, MA, USA) in Opti-MEM (Gibco, Carlsbad, CA, USA).

### 4.5. ALP Staining

Differentiated C2C12 cells in 24-well plates were washed with phosphate-buffered saline at first, and then fixed in 4% formaldehyde at room temperature for 15 min. Cells were stained with the 1-Step NBT/BCIP Substrate Solution (Thermo Fisher Scientific, Waltham, MA, USA) for 5 min and images were captured by using an automatic scanner. Quantification of ALP staining was performed by using a microplate reader at the absorbance of 480 nm.

### 4.6. Luciferase Reporter Assay

293T cells or C2C12 cells seeded in 48-well plates were transfected with CMV promoter-driven β-galactosidase (β-gal) plasmids in combination with the indicated luciferase reporter plasmids (Runx2, Osterix, and Dlx5). Luciferase Assay System (Promega, Madison, WI, USA) was used to measure the luciferase activity of indicated cell groups following the manufacturer’s protocol, and the corresponding β-gal activity were normalized to determine transfection efficiency in each group.

### 4.7. DCFH-DA Staining

Indicated groups of C2C12 cells seeded in 24-well plates were incubated with 10 μM DCFH-DA reagent (Sigma, St. Louis, MO, USA) at 37 °C for 30 min, then the fluorescence was immediately captured by using a confocal microscope at 488 nm excitation wavelength and 525 nm emission wavelength. All procedures were protected from light.

### 4.8. DHE Fluorescence Measurement

Cellular ROS level was also detected by DHE fluorescence by using DHE (Dihydroethidium) Assay Kit–Reactive Oxygen Species (Abcam, Cambridge, UK) following the manufacturer’s protocol. Indicated groups of C2C12 cells seeded in 96-well plates were incubated with 5 μM DHE at 37 °C for 30 min, and the fluorescence was measured by using a microplate reader at 495 nm excitation wavelength and 580 nm emission wavelength. All procedures were protected from light.

### 4.9. Immunofluorescence

C2C12 cells were fixed in 4% paraformaldehyde for 15 min and then permeabilized in 0.5% Triton X-100 for 15 min at room temperature. Cells were blocked by 5% BSA and incubated with an anti-MT3 (Santa Cruz Biotechnology, Santa Cruz, CA, USA, sc-164990; 1:50 dilution) antibody at 4 °C overnight, followed by an appropriate secondary antibody, and the cell nuclei were stained with DAPI. Images were visualized and captured by using a confocal microscope.

### 4.10. Statistical Analysis

Data were analyzed by using GraphPad Prism 8.0, and results were expressed as mean ± SEM. Differences in each group were evaluated by using unpaired student’s *t*-test (between two groups) or a one-way ANOVA (between multiple groups). A value of *p* < 0.05 was considered statistically significant.

## Figures and Tables

**Figure 1 ijms-22-04312-f001:**
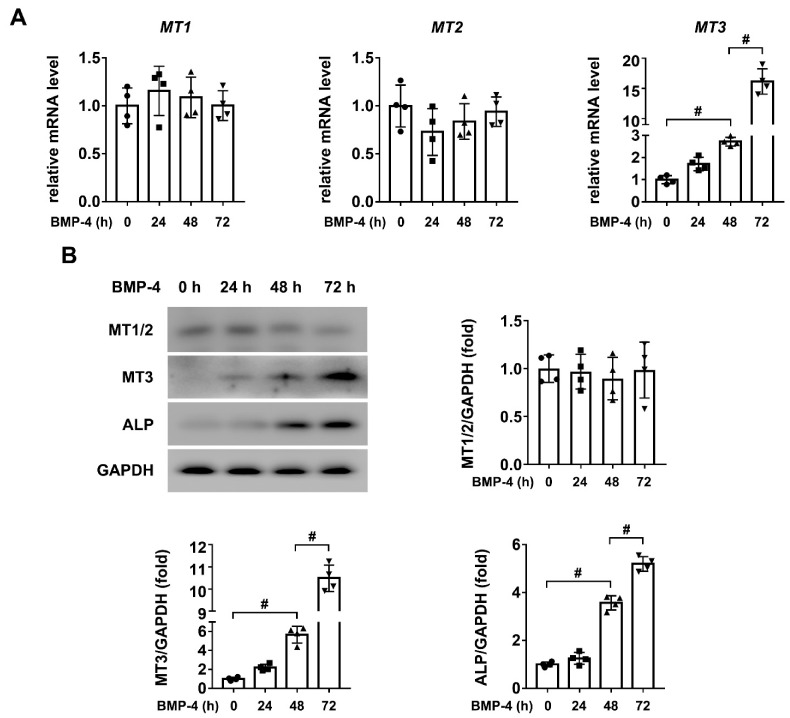
MT3 is significantly upregulated in BMP4 treated C2C12 cells. (**A**) The mRNA levels of MT1, MT2, and MT3 in C2C12 cells treated with recombinant BMP4 for 0, 24, 48, and 72 h; (**B**) Protein expression levels and quantitative analysis of MT1, MT2, MT3, and ALP in C2C12 cells treated with recombinant BMP4 for 0, 24, 48, and 72 h (GAPDH was served as the loading control). All data are presented as mean ± SEM, # *p* < 0.05.

**Figure 2 ijms-22-04312-f002:**
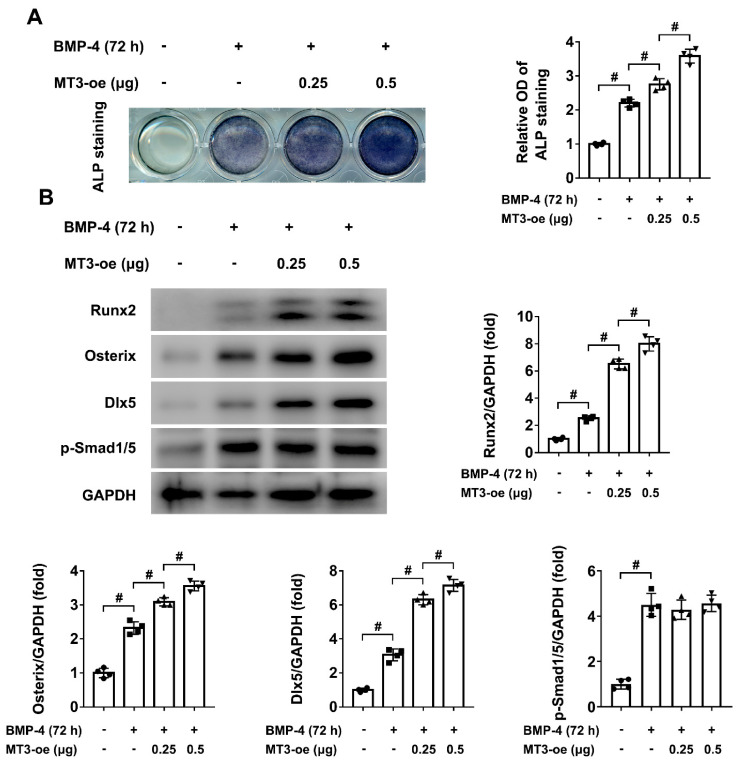
MT3 overexpression promotes BMP4-induced osteoblast differentiation in C2C12 cells. (**A**) ALP staining and quantitative analysis of ALP activity in control or MT3-overexpressing C2C12 cells treated with or without recombinant BMP4 for 72 h. (**B**) Protein expression levels and quantitative analysis of Runx2, Osterix, Dlx5, and p-Smad1/5 in control or MT3-overexpressing C2C12 cells treated with or without recombinant BMP4 for 72 h (GAPDH was served as the loading control). All data are presented as mean ± SEM, # *p* < 0.05.

**Figure 3 ijms-22-04312-f003:**
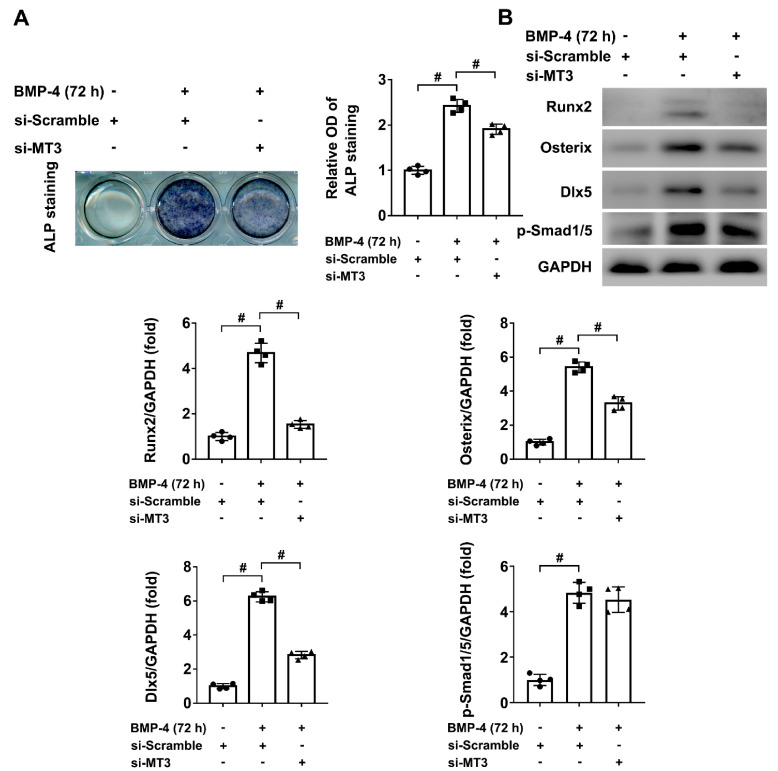
MT3 knockdown inhibited BMP4-induced osteoblast differentiation in C2C12 cells. (**A**) ALP staining and quantitative analysis of ALP activity in control or MT3 knockdown C2C12 cells treated with or without recombinant BMP4 for 72 h. (**B**) Protein expression levels and quantitative analysis of Runx2, Osterix, and Dlx5 in control or MT3 knockdown C2C12 cells treated with or without recombinant BMP4 for 72 h (GAPDH was served as the loading control). All data are presented as mean ± SEM, # *p* < 0.05.

**Figure 4 ijms-22-04312-f004:**
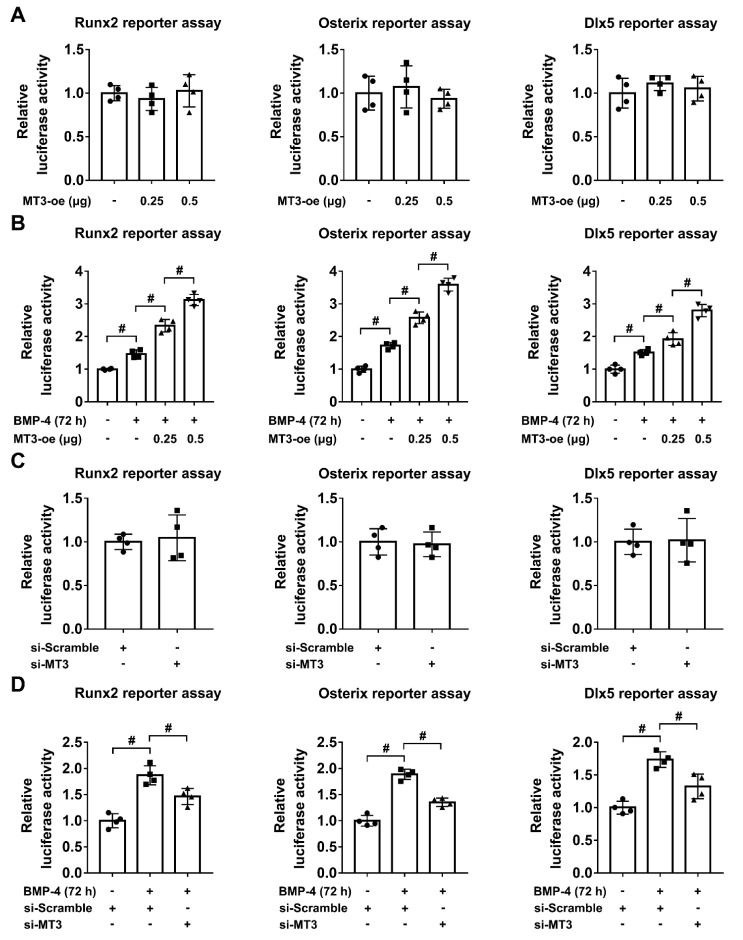
MT3 indirectly regulates Runx2/Osterix/Dlx5 activation during osteoblast differentiation. (**A**) Luciferase reporter activities of Runx2, Osterix, and Dlx5 in control or MT3-overexpressing 293T cells. (**B**) Luciferase reporter activities of Runx2, Osterix, and Dlx5 in control or MT3-overexpressing C2C12 cells treated with or without recombinant BMP4 for 72 h. (**C**) Luciferase reporter activities of Runx2, Osterix, and Dlx5 in control or MT3 knockdown 293T cells. (**D**) Luciferase reporter activities of Runx2, Osterix, and Dlx5 in control or MT3 knockdown C2C12 cells treated with or without recombinant BMP4 for 72 h. All data are presented as mean ± SEM, # *p* < 0.05.

**Figure 5 ijms-22-04312-f005:**
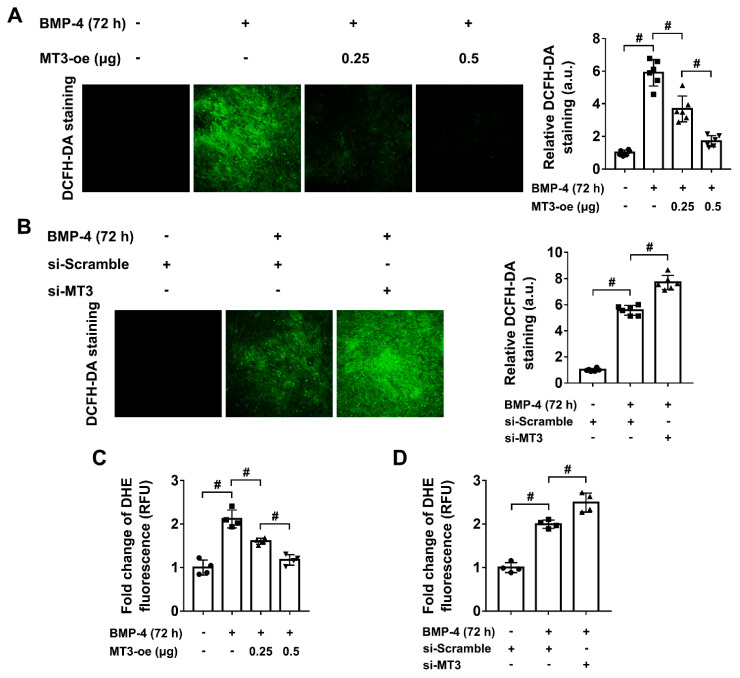
MT3 reduces oxidative stress in BMP4 treated C2C12 cells. (**A**) DCFH-DA staining and quantitative analysis of ROS production in control or MT3-overexpressing C2C12 cells treated with or without recombinant BMP4 for 72 h (magnification = 100×). Green color represents the fluorescence intensity of DCFH-DA staining. (**B**) DCFH-DA staining and quantitative analysis of ROS production in control or MT3 knockdown C2C12 cells treated with or without recombinant BMP4 for 72 h (magnification = 100×). (**C**) Quantitative analysis of DHE fluorescence in control or MT3-overexpressing C2C12 cells treated with or without recombinant BMP4 for 72 h. (**D**) Quantitative analysis of DHE fluorescence in control or MT3 knockdown C2C12 cells treated with or without recombinant BMP4 for 72 h. All data are presented as mean ± SEM, # *p* < 0.05.

**Figure 6 ijms-22-04312-f006:**
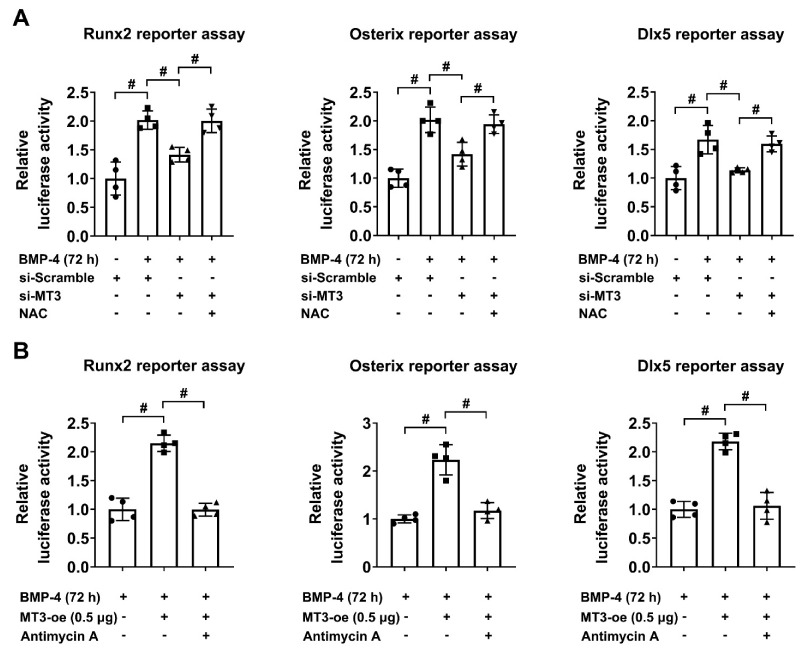
Oxidative stress affects the reporter activities of Runx2/Osterix/Dlx5 in MT3-overexpressing/silencing C2C12 cells. (**A**) Luciferase reporter activities of Runx2, Osterix, and Dlx5 in control or MT3-overexpressing C2C12 cells treated with or without recombinant BMP4 for 72 h in the absence or presence of NAC (10 mM). (**B**) Luciferase reporter activities of Runx2, Osterix, and Dlx5 in control or MT3 knockdown C2C12 cells treated with recombinant BMP4 for 72 h in the absence or presence of antimycin A (0.5 μM). All data are presented as mean ± SEM, # *p* < 0.05.

**Figure 7 ijms-22-04312-f007:**
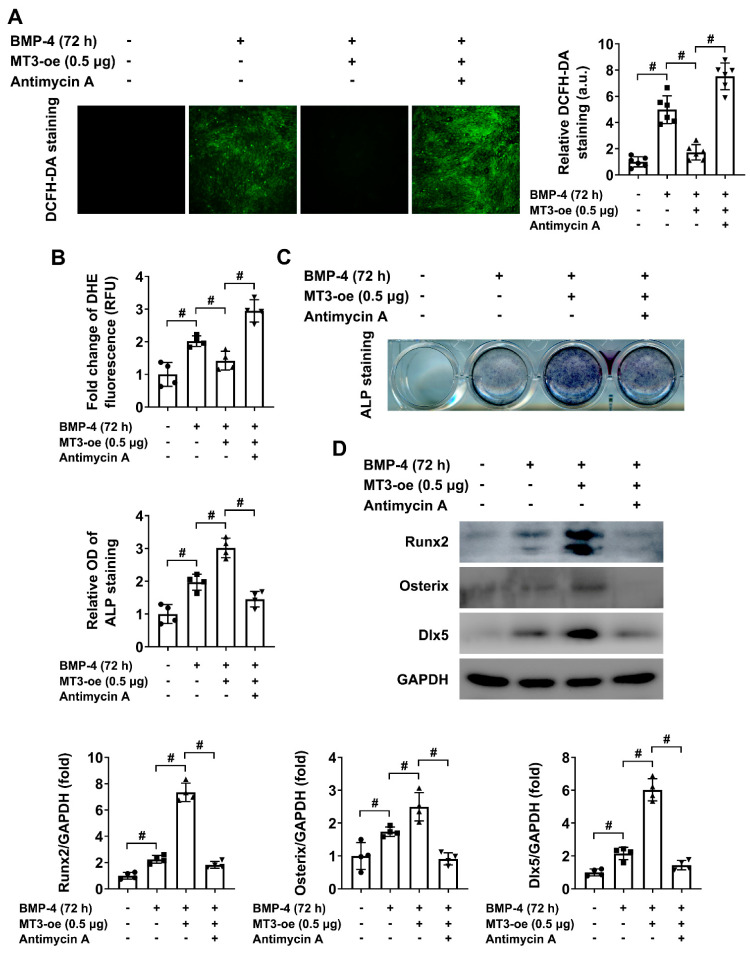
ROS production impedes BMP4-induced osteoblast differentiation in MT3-overexpressing C2C12 cells. (**A**) DCFH-DA staining and quantitative analysis of ROS production in saline- or antimycin A (0.5 μM)-handled MT3-overexpressing C2C12 cells treated with or without recombinant BMP4 for 72 h (magnification = 100×). (**B**) Quantitative analysis of DHE fluorescence in saline- or antimycin A (0.5 μM)-handled MT3-overexpressing C2C12 cells treated with or without recombinant BMP4 for 72 h. (**C**) ALP staining and quantitative analysis of ALP activity in saline- or antimycin A (0.5 μM)-handled MT3-overexpressing C2C12 cells treated with or without recombinant BMP4 for 72 h. (**D**) Protein expression levels and quantitative analysis of Runx2, Osterix, and Dlx5 in saline- or antimycin A (0.5 μM)-handled MT3-overexpressing C2C12 cells treated with or without recombinant BMP4 for 72 h. All data are presented as mean ± SEM, # *p* < 0.05.

## Data Availability

Please contact the corresponding author for reasonable data requests.

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
