# Peer review of "Metallothionein 3 Promotes Osteoblast Differentiation in C2C12 Cells via Reduction of Oxidative Stress"

_ijms, 2021, doi:10.3390/ijms22094312_

Round 1
Reviewer 1 Report
Metallothinneins (MT)s are intracellular cysteine-rich proteins whose expression is enhanced under stress condition, and they have the ability to regulate redox balance in the living organism, however, their role in regulating osteoblast differentiation is still unclear. In this research, the authors found that the expression of MT3 was specifically upregulated in the differentiation process of C2C12 myoblast cells treated with bone morphogenetic protein 4 (BMP4). Transfection with MT3-overexpressing plasmids in C2C12 cells enhanced their differentiation to osteoblasts, together with upregulating the protein expression of bone specific transcription factors runt-related gene 2 (Runx2), Osterix, and distal-less homeobox 5 (Dlx5). Additionally, MT3 knockdown had the opposite effect. Further studies revealed that overexpression of MT3 decreased reactive oxygen species (ROS) production in C2C12 cells treated with BMP4, and MT3 silencing enhanced ROS production. Treating C2C12 cells with antioxidant N-acetylcysteine also promoted osteoblast differentiation and upregulated Runx2/Osterix/Dlx5, while ROS generator antimycin A treatment had the opposite effect. Finally, antimycin A treatment inhibited osteoblast differentiation and Runx2/Osterix/Dlx5 expression in MT3-overexpressing C2C12 cells. These findings identify the role of MT3 in osteoblast differentiation and indicate that MT3 has osteogenesis promoting potential in the field of bone injury research.
In general, this manuscript is well written and provides a structural and contextual understanding for the results, as well as the therapeutic opinions. The study may deserve publication after minor revision.
- Most of the results showed a statistical significance (p<0.05) between groups. What I cannot find in the study is the number of samples in each group in order to reach a statistical significance. Otherwise the conclusion should be made more conservatively or at least, this condition should be stated in the section of the limitation of the research.
- The style of the references are incorrect, please make changes for the references to conform to the requirement of the journal.
Author Response
Metallothinneins (MT)s are intracellular cysteine-rich proteins whose expression is enhanced under stress condition, and they have the ability to regulate redox balance in the living organism, however, their role in regulating osteoblast differentiation is still unclear. In this research, the authors found that the expression of MT3 was specifically upregulated in the differentiation process of C2C12 myoblast cells treated with bone morphogenetic protein 4 (BMP4). Transfection with MT3-overexpressing plasmids in C2C12 cells enhanced their differentiation to osteoblasts, together with upregulating the protein expression of bone specific transcription factors runt-related gene 2 (Runx2), Osterix, and distal-less homeobox 5 (Dlx5). Additionally, MT3 knockdown had the opposite effect. Further studies revealed that overexpression of MT3 decreased reactive oxygen species (ROS) production in C2C12 cells treated with BMP4, and MT3 silencing enhanced ROS production. Treating C2C12 cells with antioxidant N-acetylcysteine also promoted osteoblast differentiation and upregulated Runx2/Osterix/Dlx5, while ROS generator antimycin A treatment had the opposite effect. Finally, antimycin A treatment inhibited osteoblast differentiation and Runx2/Osterix/Dlx5 expression in MT3-overexpressing C2C12 cells. These findings identify the role of MT3 in osteoblast differentiation and indicate that MT3 has osteogenesis promoting potential in the field of bone injury research.
In general, this manuscript is well written and provides a structural and contextual understanding for the results, as well as the therapeutic opinions. The study may deserve publication after minor revision.
Response: Thank you very much for your kind help and valuable comments to our manuscript, we believe that these helpful suggestions will highly improve the quality of our manuscript, and have done the revision accordingly.
- Most of the results showed a statistical significance (p<0.05) between groups. What I cannot find in the study is the number of samples in each group in order to reach a statistical significance. Otherwise the conclusion should be made more conservatively or at least, this condition should be stated in the section of the limitation of the research.
Response: Thank you for the suggestion, in the discussion part of the revised manuscript, we have stated that the small sample size of our study is a limitation of the research according to your comment (page 11-12, line 307-310).
- The style of the references are incorrect, please make changes for the references to conform to the requirement of the journal.
Response: Thank you for your comment, we have modified the reference style of our manuscript strictly following the journal’s requirement now.
Reviewer 2 Report
The paper is well written and the topic represents an important concern about biology of bone healing and injuries treatment. Indeed, results presented by authors can lead to new applications of the proposed protocol to improve techniques for bone regeneration in vivo, also in the light of potential clinical application on dental patients. the materials and methods are adequate and rigorous, results are clearly described and fit very well with proposed objectives.
The weakness of the paper is linked to the introduction and discussion, where Authors should improve some sentences about potential link with clinical topics and potential clinical application. Indeed, they should include in their references some papers about the following crucial clinical points. First of all, which is the role of metallothionein 3 in the cells (e.g. osteoblasts) in an area reconstructed or treated with biomaterials (PubMed ID: 21841997; PubMed ID: 27496576; PubMed ID: 22010090; PubMed ID: 28108959) or in areas treated with sinus lifting (PubMed ID: 21599829; PubMed ID: 23057028; PubMed ID: 20977614)
Another very important topic that Authors should discuss is the role of metallothionein 3 for osteoblast in infections condition: a potential diminished rate of bacterial colonization/management could significantly improve the success and survival of implant-prosthetic rehabilitations in immunocompromised patients (PubMed ID: 25955953; PubMed ID: 26238779;PubMed ID: 31781702), also in endodontics (PubMed ID: 18811596), avoiding facial perimandibular abscesses (PubMed ID: 19821124)
After these changes the paper can be accepted for publication
Author Response
The paper is well written and the topic represents an important concern about biology of bone healing and injuries treatment. Indeed, results presented by authors can lead to new applications of the proposed protocol to improve techniques for bone regeneration in vivo, also in the light of potential clinical application on dental patients. the materials and methods are adequate and rigorous, results are clearly described and fit very well with proposed objectives.
The weakness of the paper is linked to the introduction and discussion, where Authors should improve some sentences about potential link with clinical topics and potential clinical application. Indeed, they should include in their references some papers about the following crucial clinical points. First of all, which is the role of metallothionein 3 in the cells (e.g. osteoblasts) in an area reconstructed or treated with biomaterials (PubMed ID: 21841997; PubMed ID: 27496576; PubMed ID: 22010090; PubMed ID: 28108959) or in areas treated with sinus lifting (PubMed ID: 21599829; PubMed ID: 23057028; PubMed ID: 20977614)
Another very important topic that Authors should discuss is the role of metallothionein 3 for osteoblast in infections condition: a potential diminished rate of bacterial colonization/management could significantly improve the success and survival of implant-prosthetic rehabilitations in immunocompromised patients (PubMed ID: 25955953; PubMed ID: 26238779;PubMed ID: 31781702), also in endodontics (PubMed ID: 18811596), avoiding facial perimandibular abscesses (PubMed ID: 19821124)
After these changes the paper can be accepted for publication
Response: Thank you very much for the helpful and valuable suggestions to our manuscript, we believe that these comments will highly improve the quality of the introduction and discussion part of our paper, therefore, we have cited all of the indicated references and other necessary references together with appropriate sentences in the introduction (page 2, line 57-62, reference No. 11-17) and discussion (page 11, line 284-293, reference No. 43-49) part of the revised manuscript now.
Reviewer 3 Report
The authors investigated the role of metallothioneins (MTs) on BMP4-induced osteoblastic differentiation using C2C12 cells. Among MTs, only MT3 was upregulated during osteoblastic differentiation from C2C12 cells, induced by BMP4. The overexpression of MT3 enhanced BMP4-induced osteoblastic differentiation by upregulating RUNX2, Osterix, and Dlx expressions, while knockdown of MT3 using siRNA suppressed these effects. MT3 suppressed reactive oxygen species (ROS) production in C2C12 cells treated with BMP4, and knockdown of MT3 canceled these effects. The treatment with antioxidant N-acetylcysteine (NAC) also enhanced BMP4-induced osteoblastic differentiation. On the other hand, ROS generator antimycin A treatment inhibited osteoblastic differentiation in MT3-overexpressing C2C12 cells induced by BMP4. The authors concluded that MT3 has osteogenesis promoting potential in the field of bone injury research. To support the authors’ conclusion, several additional experiments are needed. Since there are many mistakes in English throughout the manuscript, please check with a native speaker before submission.
- Abstract; “Metallothinneins” should be “Metallothioneins”.
- Fig.1B; The authors should examine the changes in not only MT3, but also MT1 and MT2 by Western blot.
- Fig. 1; Is the increased expression of MT3 a direct effect of BMP stimulation? Check if the BMP responsive element is present in the MT3 promoter.
- Figs 2 and 3; The authors should show the transfection efficiency using immunostainig with anti-MT3 antibody.
- Figs 2 and 3; The authors should examine the changes in p-Smad1/5 by Western blot.
- Supplemental Figure; The authors should show the changes in MT3 protein using anti-MT3 antibody.
- Fig. 3B; The author should add MT3 expression.
- Figs 5A,B, 6A, and 7A; The images are poor. The authors should replace better images.
- Fig. 5; The authors should examine whether NAC restore the inhibitory effects of the knockdown of MT3 on the transcriptional activity of Runx2, Osterix, and Dlx.
- Fig. 6; Since this Figure shows that NAC promotes the effect of BMP4 without MT3, it is not necessary to show.
- Fig. 7; Control (vehicle) is required as a negative control, and BMP4 treatment is required as a positive control.
- The authors should examine whether Antimycin suppress the effects of the overexpression of MT3 on the transcriptional activity of Runx2, Osterix, and Dlx.
- Discussion; The authors should cite the following paper and consider the differences in experimental results. Mandal CC, et al. Biochem J. 2011;433(2):393-402.
- Unless in vivo experiments, the authors cannot conclude that MT3 has osteogenesis promoting potential in the field of bone injury research.
Author Response
The authors investigated the role of metallothioneins (MTs) on BMP4-induced osteoblastic differentiation using C2C12 cells. Among MTs, only MT3 was upregulated during osteoblastic differentiation from C2C12 cells, induced by BMP4. The overexpression of MT3 enhanced BMP4-induced osteoblastic differentiation by upregulating RUNX2, Osterix, and Dlx expressions, while knockdown of MT3 using siRNA suppressed these effects. MT3 suppressed reactive oxygen species (ROS) production in C2C12 cells treated with BMP4, and knockdown of MT3 canceled these effects. The treatment with antioxidant N-acetylcysteine (NAC) also enhanced BMP4-induced osteoblastic differentiation. On the other hand, ROS generator antimycin A treatment inhibited osteoblastic differentiation in MT3-overexpressing C2C12 cells induced by BMP4. The authors concluded that MT3 has osteogenesis promoting potential in the field of bone injury research. To support the authors’ conclusion, several additional experiments are needed. Since there are many mistakes in English throughout the manuscript, please check with a native speaker before submission.
Response: Thank you very much for all of the constructive and helpful comments to our manuscript, we believe that all of these valuable suggestions can highly improve the quality of our research, for your concern of the weakness of the English quality throughout the manuscript, we have now asked a native English speaker to carefully check and edit our manuscript, and we have also tried our best to improve the English level in the revised manuscript.
- Abstract; “Metallothinneins” should be “Metallothioneins”.
Response: Thank you for your help, this has been corrected in the revised manuscript (page 1, line 14).
- Fig.1B; The authors should examine the changes in not only MT3, but also MT1 and MT2 by Western blot.
Response: Thank you for the comment, we have added the result of MT1 and MT2 expression in revised Figure. 1B by using a specific antibody and added appropriate words in the results part of the revised manuscript (page 3, line 107).
- Fig. 1; Is the increased expression of MT3 a direct effect of BMP stimulation? Check if the BMP responsive element is present in the MT3 promoter.
Response: Thank you for your suggestion, this would be an excellent experiment and will broaden the understanding of the relationship between MT3 and BMP signaling, the indicated experiment is actually in progress but unfortunately, as we do not have any plasmids containing BMP responsive element or MT3 promoter, we have to start to construct new plasmids in our lab, therefore we cannot finish this experiment in a short period, and we also cannot guarantee that the experiment could be done or get good results in the due time of the revision. Therefore, we have to added this as a limitation of our study in the discussion part of the revised manuscript (page 11, line 302-305). We apologize for the unfinished work.
- Figs 2 and 3; The authors should show the transfection efficiency using immunostainig with anti-MT3 antibody.
Response: Thank you for your suggestion, transfection efficiency has been validated by using anti-MT3 antibody accordingly, results are represented in revised Supplementary Figure. 1A&B.
- Figs 2 and 3; The authors should examine the changes in p-Smad1/5 by Western blot.
Response: Thank you for your suggestion, indicated experiments has been added and results are represented in revised Figure. 2B and Figure. 3B. Appropriate sentences were also added in the results part of the revised manuscript (page 4, line 123-126 and page 5, line 140-141).
- Supplemental Figure; The authors should show the changes in MT3 protein using anti-MT3 antibody.
Response: Thank you for your comment, results are represented in revised Supplementary Figure. 1A&B.
- Fig. 3B; The author should add MT3 expression.
Response: Thank you for your comment, similarly, results are represented in revised Supplementary Figure. 1A&B.
- Figs 5A,B, 6A, and 7A; The images are poor. The authors should replace better images.
Response: Thank you for your suggestion, accordingly, all the DCFH-DA staining result throughout the manuscript were checked (containing original Figure. 5A&B, Figure. 6A, Figure. 7A, and Figure. 8A), and high-resolution original pictures were used to replace the older images in the indicated figures (now are the revised Figure. 5A&B, Figure. 7A, Supplementary Figure. 2A, and Supplementary Figure. 3A).
- Fig. 5; The authors should examine whether NAC restore the inhibitory effects of the knockdown of MT3 on the transcriptional activity of Runx2, Osterix, and Dlx.
Response: Thank you for your suggestion, indicated experiments and appropriate words have been added in the revised manuscript (revised Figure. 6A, page 8, line 201-204).
- Fig. 6; Since this Figure shows that NAC promotes the effect of BMP4 without MT3, it is not necessary to show.
Response: Thank you for your comment, accordingly, the original Figure. 6 and the associated original Figure. 7 were both deleted from the main file of the manuscript and moved to the supplementary materials of the revised manuscript, we did not completely delete these two figures from the manuscript because we think that response to the No. 13 comment still need these results.
- Fig. 7; Control (vehicle) is required as a negative control, and BMP4 treatment is required as a positive control.
Response: Thank you for your comment, the original Figure. 7 (now is the revised Supplementary Figure. 3) has contained vehicle and BMP4 treatment alone.
- The authors should examine whether Antimycin suppress the effects of the overexpression of MT3 on the transcriptional activity of Runx2, Osterix, and Dlx.
Response: Thank you for your suggestion, indicated experiments and appropriate words have been added in the revised manuscript (revised Figure. 6B, page 8, line 204-208).
- Discussion; The authors should cite the following paper and consider the differences in experimental results. Mandal CC, et al. Biochem J. 2011;433(2):393-402.
Response: Thank you for your comment, in the discussion part of the revised manuscript, we have cited the indicated reference and added appropriate sentences (page 11, line 267-272).
- Unless in vivo experiments, the authors cannot conclude that MT3 has osteogenesis promoting potential in the field of bone injury research.
Response: Thank you for your suggestion, we have checked and deleted the indicated sentences and replaced them with other appropriate words accordingly (page 1, line 29 and page 11, line 264).
Round 2
Reviewer 3 Report
I appreciate your effort in trying to address the reviewers' questions and comments. Nonetheless, the manuscript has some deficiencies and the following concerns need to be fully addressed.
- Fig. 1; It is possible to check the presence of BMP responsive elements in the promoter region of MT3 in public databases without a reporter assay.
- Figs. 2 and 3; The reviewer mentioned that the authors should show the transfection efficiency using “IMMUNOSTAINING” but NOT immunoblot with anti-MT3 antibody. The reviewer has already noticed that you have shown the changes of expression level of MT3 in Fig S1. My concern is that the authors performed experiments with transient transfection without establishing a stable cell line. If the transfection efficiency is around 10%, why does the whole cell stain blue with ALP staining? The cells were stained with MT3 and instructed to show the ratio of MT3 expression to the whole cells using nuclear staining. This is the same concern is for siRNA experiments.
- 3. Fig. 7; Control (vehicle) is required as a negative control, and BMP4 treatment is required as a positive control. It is necessary to show the data how much the MT3 promoting effect and the inhibitor suppressing effect are compared with the positive and negative controls in the same experimental group. Because, there are variations in each experiment.
Author Response
I appreciate your effort in trying to address the reviewers' questions and comments. Nonetheless, the manuscript has some deficiencies and the following concerns need to be fully addressed.
Response: Thank you for your kind help and valuable suggestions to our manuscript, we believe that those constructive suggestions will greatly improve the quality of our paper, and we have tried our best to fully address all of the concerns in this submission.
Fig. 1; It is possible to check the presence of BMP responsive elements in the promoter region of MT3 in public databases without a reporter assay.
Response: Thank you for your valuable suggestion, we have checked previous publications and we found that Sp1 increases the promoter activity of MT3, while BMP4 increases the expression of Osterix (Sp7) and thus affecting Sp1, this may explain why BMP4 treatment upregulates MT3, we also added those discussions in the revised manuscript (page 12, line 296-302). In this submission we did not search public promoter region databases as our lab lacks relevant experiences in this field.
Figs. 2 and 3; The reviewer mentioned that the authors should show the transfection efficiency using “IMMUNOSTAINING” but NOT immunoblot with anti-MT3 antibody. The reviewer has already noticed that you have shown the changes of expression level of MT3 in Fig S1. My concern is that the authors performed experiments with transient transfection without establishing a stable cell line. If the transfection efficiency is around 10%, why does the whole cell stain blue with ALP staining? The cells were stained with MT3 and instructed to show the ratio of MT3 expression to the whole cells using nuclear staining. This is the same concern is for siRNA experiments.
Response: Thank you for your valuable suggestion, we have used immunofluorescence staining of MT3 in the indicated figures now (revised Supplementary Figure. 1B).
- Fig. 7; Control (vehicle) is required as a negative control, and BMP4 treatment is required as a positive control. It is necessary to show the data how much the MT3 promoting effect and the inhibitor suppressing effect are compared with the positive and negative controls in the same experimental group. Because, there are variations in each experiment.
Response: Thank you for your valuable suggestion, the original Figure 7 has been carefully revised according to your comment now (revised Figure. 7A-D).
